# Comparison of Structured Nutrition Therapy for Ramadan with Standard Care in Type 2 Diabetes Patients

**DOI:** 10.3390/nu12030813

**Published:** 2020-03-19

**Authors:** Barakatun-Nisak Mohd Yusof, Wan Zul Haikal Hafiz Wan Zukiman, Zalina Abu Zaid, Noraida Omar, Firdaus Mukhtar, Nor Farahain Yahya, Aainaa Syarfa Mohd Shahar, Farah Yasmin Hasbullah, Rachel Liu Xin Yi, Agnieszka Marczewska, Osama Hamdy

**Affiliations:** 1Department of Nutrition & Dietetics, Faculty of Medicine & Health Sciences, Universiti Putra Malaysia, Serdang 43400, Selangor, Malaysia; zalina@upm.edu.my (Z.A.Z.); noraidaomar@upm.edu.my (N.O.); farahainyahya93@gmail.com (N.F.Y.); ainasyarfa@gmail.com (A.S.M.S.); farahyasmin90@gmail.com (F.Y.H.); 2Institute for Social Science Studies, Universiti Putra Malaysia, Serdang 43400, Selangor, Malaysia; drfirdaus@upm.edu.my; 3Department of Medicine, Faculty of Medicine & Health Sciences, Universiti Putra Malaysia, Serdang 43400, Selangor, Malaysia; zulhaikal@upm.edu.my; 4Department of Psychiatry, Faculty of Medicine & Health Sciences, Universiti Putra Malaysia, Serdang 43400, Selangor, Malaysia; 5Nestlé Health Science, Petaling Jaya 47810, Malaysia; Rachel.LiuXinYi@my.nestle.com; 6Nestlé Health Science, 1800 Vevey, Switzerland; Agnieszka.Marczewska@nestle.com; 7Joslin Diabetes Centre, Harvard Medical School, MA 02215, USA; Osama.Hamdy@joslin.harvard.edu

**Keywords:** structured nutrition therapy, Ramadan nutrition plan, type 2 diabetes, diabetes-specific formula, HbA1c, fasting

## Abstract

(1) Background: Structured nutrition therapy (NT) is essential for the management of type 2 diabetes (T2D), but the optimal delivery during Ramadan fasting remains unclear. The present study aimed to evaluate the effect of structured NT program versus standard care in patients with T2D during Ramadan. (2) Methods: The present study was an 8-week, parallel, non-randomized study with patients’ preference design involving 64 patients with T2D. The participants were asked to choose their preferred group, i.e., structured NT (Structured Ramadan NT, sRNT) or standard care (SC). The participants in the sRNT group received a Ramadan-focused nutrition plan, including a diabetes-specific formula throughout the study, whereas the patients in the SC group received standard nutrition care. Study outcomes included clinical outcomes and quality of life (QoL). Data was analyzed using two-way repeated-measures ANOVA and linear mixed-effects model. (3) Results: More than half of the participants (*n* = 38, 63%) chose sRNT as their preferred group. Both groups had comparable baseline characteristics. After 8-weeks of the respective intervention, participants in the sRNT group had lower levels of fasting plasma glucose (−0.9 ± 0.3 mmol/L vs. 0.2 ± 0.3 mmol/L, *p* < 0.05), triglycerides (−0.21 ± 0.08 mmol/L vs. 0.20 ± 0.17 mmol/L, *p* < 0.05), and self-monitoring glucose at pre-dawn (6.9 mmol/L vs. 7.8 mmol/L, *p* < 0.05) and pre-bedtime (7.6 mmol/L vs. 8.6 mmol/L, *p* < 0.05) than participants in the SC group. Although not different between groups, HbA1c levels decreased significantly in the sRNT (−0.72 ± 0.16%, *p* < 0.001) but not in the SC group (−0.35 ± 0.24%, *p* = 0.155). QoL and satisfaction scores improved significantly in sRNT group, but not in SC group. (4) Conclusions: The structured NT regimen for Ramadan is a feasible and beneficial program for T2D patients observing Ramadan fasting as it showed an improvement in clinical outcomes and QoL.

## 1. Introduction

Daytime fasting during the Ramadan month is an important religious practice in Islam. During the entire month of Ramadan, all healthy Muslims fast every day from dawn to sunset with fasting hours across the world varying between 11 and 22 hours, depending on the latitude. In Southeast Asia, including Malaysia, the average fasting period lasts about 14 hours. During the daytime in Ramadan, people abstain from any food or beverage, including water. The daily diet consists of two main meals named as *Suhoor*, consumed before dawn, and *Iftar*, consumed after sunset. The daily Ramadan routine causes a sudden change in meal and sleep patterns, which increases the risk of nutritional issues [1]. 

The shift in eating patterns has significant implications for the physiology process, in particular to people with diabetes. When fasting, insulin resistance or deficiency state can cause excessive glycogen breakdown leading to gluconeogenesis. The alteration put individuals with diabetes at an increased risk to develop hypoglycemia, hyperglycemia, ketoacidosis, and dehydration [2]. Therefore, they are exempted from fasting, but many Muslim adults with diabetes continue to observe Ramadan fasting, even against medical advice. The CREED study documented that 94% of patients with type 2 diabetes (T2D) fasted for at least 15 days during Ramadan, with more than half (64%) fasting every day [3]. Nonetheless, with proper Ramadan-focused education, medical risk, including hypoglycemia, can be minimized [4]. 

In diabetes, changes in timing and meal composition have a considerable impact on glycemic control [5]. Long hours of abstaining from foods during the daytime may lead to excessive food consumption at night, comprising vast amounts of food rich in carbohydrates. The International Diabetes Federation and the Diabetes and Ramadan International Alliance have recently published the guideline for Ramadan Nutrition Plan (RNP) [6]. No study has determined the feasibility of RNP in the real clinical scenario; therefore, further investigation to elucidate the effects of RNP is warranted. The effects of structured nutrition therapy (NT), including the use of diabetes-specific formula (DSF) to facilitate weight management and glycemic control in T2D, have been well established [7,8,9], but these studies have been performed outside the Ramadan period. The optimal administration of structured NT regimen during Ramadan fasting remains unclear. Many studies have reported the benefits of providing Ramadan-focused intervention [4,10,11,12,13,14]. Nonetheless, none of these studies have explored the potential of structured NT, including the use of DSF in patients’ diets during Ramadan fasting. The best criterion for inclusion of DSF during Ramadan is uncertain. Therefore, the present study aimed to evaluate the effect of structured NT versus standard care during Ramadan fasting on clinical parameters and quality of life (QoL) in patients with T2D.

## 2. Materials and Methods 

### 2.1. Study Population

Muslim patients aged 18–65 years, who had a confirmed diagnosis of T2D for at least 3 months and expressed their intention to fast for at least 15 days during Ramadan, were recruited in the study. They were not on insulin therapy and treated with a stable dose of other diabetes medications for >2 months before enrolment. They had their HbA1c level between 6.5% and 12.0%, and body mass index (BMI) between 18.5 to 40.0 kg/m^2^. They were also adults who were able to perform activities of daily living (ADLs) independently, such as cooking and preparing meals themselves. The following patients were excluded from the study: pregnant or breastfeeding women, patients with a history of hypoglycemia leading to hospitalization during the previous Ramadan fasting, and the patients who were actively enrolled in weight management programs. Study participants were the patients who visited the Universiti Putra Malaysia Health Centre for their routine diabetes treatment and were recruited through advertisement and clinic referrals. After screening, eligible participants signed the study consent form. The Institutional Review Board approved the study (reference number JKEUPM-2019-009) with the trial registration at clinicaltrials.gov (identifier: NCT03817099) and Malaysia National Medical Research Register (registration number 45868).

### 2.2. Study Design 

The present study was an 8-week, parallel-group, non-randomized study conducted between April 2019 and July 2019. The design considered patients’ preference for the allocation to a study arm. Eligible participants chose their preferred group, i.e., structured NT (structured Ramadan Nutrition Therapy, sRNT) or standard care group (SC). The primary outcomes included patients’ preference for the study groups, and glycemic parameters, including fasting plasma glucose, HbA1c, and self-monitoring blood glucose profiles. Secondary outcomes included changes in body weight, cardiovascular risk factors, dietary intake, and QoL. A sub-sample of the participants (*n* = 21) were invited to participate in the continuous glucose monitoring (CGM) study to evaluate the glucose profiles during Ramadan fasting further.

### 2.3. Study Procedures

#### 2.3.1. Structured Ramadan Nutrition Therapy Group

Participants in the sRNT group received the RNP according to the International Diabetes Federation and the Diabetes and Ramadan International Alliance guidelines [6], including the use of DSF after consulting a research dietician. Participants visited the health clinic four times: at baseline (1–2 weeks before Ramadan), pre-Ramadan, end-Ramadan, and post-Ramadan (2–3 weeks after end-Ramadan) for a total period of 8 weeks (Figure 1). The meal plans aimed to meet daily caloric targets either for weight reduction or maintenance. A dietary plan for a reduced caloric intake of 1200–1500 kcal/day for women and 1500–1800 kcal/day for men was designed and distributed between the two main meals: *Suhoor* (pre-dawn meal) and *Iftar* (sunset meal), and two snacks (if necessary) (Appendix A). In particular, dietary carbohydrates (~50% of total energy intake (TEI)) were distributed accordingly among the four mealtimes (Appendix A).

The structured NT recommends the use of DSF within participants’ caloric and carbohydrate limits [9]. Unlike other studies, where the DSF has been used as a meal replacement [7,8,9], our study incorporated at least one serving of the DSF (Nutren Untuk Diabetik^®^, Nestlé Health Science, Switzerland) per day during Ramadan and to be consumed together with other foods at *Suhoor* (pre-dawn meal) and/or at a pre-bed snack (Snack 2) within caloric and carbohydrate limits. The DSF provides approximately 250 kcal per serving with balanced macronutrients composition. The DSF contains prebiotics, is rich in dietary fibre, high in whey protein, and low in glycemic index (GI) (Nutren Untuk Diabetik^®^, Nestlé Health Science, Switzerland).

Participants received a structured RNP using a specific Ramadan toolkit consisting of a flip chart on Ramadan-focused education and construction of the balanced meal plan for Ramadan. They were provided with a 14-day meal plan based on the Malaysian diet (Appendix A) and Ramadan nutrition plate (Appendix A). With this plate, participants were taught the portion size using hand jive [15] and the meal order method of eating vegetables before carbohydrates [16]. During the study, nutritional issues were discussed and addressed by a research dietician via social networking applications. In the final week of Ramadan, participants also received a nutrition plan for the festive season (*Syawal* nutrition plan). Table 1 summarizes the nutrition strategies in sRNT.

#### 2.3.2. Standard Care Group

Participants in the SC group visited the health clinic three times, i.e., at baseline, end-Ramadan, and post-Ramadan during the 8 weeks study period (Figure 1). The patients in the SC group continued to follow the standard of nutrition care recommendations (Table 1) [17]. The participants’ usual intake was modified by a research dietician to suit Ramadan fasting. The portion size of the meals was demonstrated using a Healthy Malaysian plate [18]. This method teaches the individual to plan meals such that half of their plate is filled with non-starchy vegetables, and the remainder of the plate is divided between lean protein and carbohydrate. During the study, nutritional issues were discussed and addressed within the frame of standard nutrition care. 

### 2.4. Outcome Measurements 

Participants’ preferences for each study group were assessed at the end of the study. Other feasibility outcomes were measured at three time points in both groups, i.e., at baseline, end-Ramadan, and post-Ramadan during a total study duration of 8 weeks. Anthropometry measurements, blood pressure, and blood samples were assessed at each visit. Body weight was measured using a calibrated scale (TANITA HD-319, Tanita Corporation, Tokyo, Japan), height was measured using a stadiometer (SECA, Hamburg, Germany), and patients’ body mass index (BMI) was calculated. Waist circumference was measured at the midpoint between the lower margin of the palpable rib and the top of the iliac crest with a non-elastic measuring tape (SECA 201, SECA, Hamburg, Germany). 

Seated blood pressure was measured twice using an automated device (OMRON HEM-7120, Omron, Kyoto, Japan) after a 5 min rest. Before Ramadan, blood was withdrawn after an overnight fast of 10 h, while during Ramadan, blood was taken 10 h after *Suhoor* and was analyzed using an automated system in an established laboratory (B.P. Clinical Lab Sdn Bhd, Shah Alam, Malaysia). Blood measurements included the estimation of levels of HbA1c, fasting blood glucose, and lipid profile. 

Dietary intake and macronutrients were assessed using 3-day food records. Participants were taught, and written instructions were given for recording the food intake. The research dietician reviewed food records at each visit. The average of 3-day food intake data was analyzed for daily total energy intake (TEI), the proportion of macronutrient composition from TEI (%TEI), total fiber and selected micronutrient intake using the Nutritionist Pro software (V.5.1.0, Axxya Systems, WA, USA). The reliability of food intake data was ensured by including only those with plausible energy intake of 400–3500 kcal/day for women, and 700–4000 kcal/day for men [19,20]. The assessment of under-reporting of energy intake was performed at baseline using the Goldberg method [21].

QoL was measured at baseline and post-Ramadan visits using the Malay version of the Revised version of Diabetes Quality of Life (RV-DQOL13) questionnaire for adult population with Type 2 Diabetes Mellitus [22]. The RV-DQOL13 consists of 13 items, measuring three domains: (i) satisfaction, (ii) impact, and (iii) worry. Reliability was confirmed with the Cronbach’s alpha coefficients of 0.846 and 0.941 for total and the three domains, respectively [22]. 

All participants were requested to record a daily finger-pricked capillary self-monitoring blood glucose (SMBG) level using provided glucometer (Contour Plus One, Ascensia Diabetes Care, Basel, Switzerland) at five-time points throughout the study period. During the non-Ramadan period, the time points of daily assessments were at fasting (between 5 and 7 am), 2-h post-breakfast (between 7 and 9 am), pre-lunch (between 11 am and 1 pm), pre-dinner (between 6 and 8 pm), and pre-bed (between 10 and 11 pm). During Ramadan, daily blood glucose was monitored at pre-dawn (between 4 and 5 am), 2-h post-dawn meal (between 6 and 7 am), at midday (between 12 noon and 1 pm), pre-*Iftar* (between 6 and 7 pm) and pre-bed (between 10 and 11 pm). 

To further understand the glucose profiles during Ramadan fasting, a sub-sample of the participants (*n* = 21) volunteered to join the CGM (Medtronic Minimed^®^, Northridge, CA, USA) study. CGM sensor was inserted into the fatty areas of the abdomen or gluteus (approximately 2 inches above the navel or below the waistline) while avoiding areas where the body naturally bends. For validity purposes, participants performed finger prick calibrations using the glucometer up to five times a day. After 5 days, the sensor was removed, and data was downloaded to a given software (Medtronic CareLink iPro, Northridge, CA, USA). The outcome measures included average sensor glucose, estimated HbA1c, percentage of time-within-target (3.9–8.3 mmol/L), time-below-target (<3.9 mmol/L), and time-above-target (>8.3 mmol/L) glucose ranges per day [23].

### 2.5. Sample Size 

A total of 70 participants was sufficient to detect a mean difference of 0.84% in HbA1c level among T2DM patients after following a telemonitoring intervention during Ramadan with 80% power and 90% confidence interval [24]. A total of 84 participants would be needed in the study after considering a dropout rate of 20%. In this feasibility study, we recruited at least 70% of participants from the primary planned statistical analyses [25]. Hence, a minimum of 59 subjects were required for this study.

### 2.6. Statistical Analyses 

Statistical analysis was performed using the latest SPSS version for Windows (IBM Corp, Armonk, NY, USA; 2013); *p*-value < 0.05 was considered as statistically significant. Data are expressed as mean ± standard deviation (SD) for continuous parameters and percentages for categorical parameters. The normality of the data was confirmed using the Shapiro–Wilk test. Baseline characteristics of the two groups were compared using an independent t-test for continuous variables and Pearson’s chi-squared test for categorical variables. The changes over time were analyzed using a two-way repeated-measures ANOVA considering the effect of time, group, and the interaction effects (intervention effects) as the main study outcomes. The primary analysis was intention-to-treat (ITT) as it provides an unbiased estimate of treatment effect [26]. Using the ITT, all participants were included in the analysis regardless of drop-out and adherence to study protocol. Imputation of missing data using the last observation carried forward produced no effect on the significance in per-protocol (PP) analysis. Hence, we present the results for the planned ITT analysis. The SMBG data were analyzed using the linear mixed-effects model, which provides a flexible, likelihood-based approach to treat missing data and within-subject correlation. For CGM data, ANOVA was used for analysis after adjusting for the baseline value. 

## 3. Results

### 3.1. Recruitment and Baseline Characteristics

We identified 138 individuals for eligibility, among whom 64 agreed to participate in the study (Figure 2). In total, 60 participants completed the study (6% attrition rate) with more than half (*n* = 38, 63%) of the participants choosing the sRNT regimen, and the remaining (*n* = 22; 37%) continuing to receive the standard care (SC group, Figure 2). The dropout rate was comparable between the two groups. The leading causes of dropout included time constraints and uncontactable for education. Three participants in the sRNT group declined to participate before receiving the intervention, and one participant from the SC group dropped out during Ramadan’s visit (Figure 2). 

Both groups were comparable at baseline (Table 2). The study included 31 men and 33 women with a mean age of 48.3 ± 9.4 years. At baseline, study participants had a mean HbA1c of 8.0 ± 1.5% and mean diabetes duration of 5.2 ± 4.1 years (Table 2). Their initial BMI was 30.3 ± 5.4 kg/m^2^, with 87.5% of the participants being overweight or obese. Almost half (48.4%) of the participants had previously met a dietician, but only a small proportion (17.2%) had received specific Ramadan advice. During the previous Ramadan, a small number of participants experienced hypoglycemia, but the difference was not significant (*n* = 4, 9.8% in sRNT vs. *n* = 1, 4.3% in SC, *p* = 0.312). The sRNT group had more women (58.5%), and the participants had a longer duration of diabetes (6 ± 4. years), and more comorbidities (0.9 ± 0.8) than the participants in SC group. The sRNT participants (47.5%) were mainly treated with a sulphonylurea, whereas the majority of SC participants (52.2%) received metformin (Table 2).

### 3.2. Clinical Outcomes

Participants in the sRNT group showed significantly bigger reduction in fasting blood glucose at end-Ramadan (−1.47 ± 0.30 mmol/L vs. −0.25 ± 0.40 mmol/L, *p* < 0.05) and post-Ramadan (−0.90 ± 0.32 mmol/L vs. 0.15 ± 0.30 mmol/L, *p* < 0.05) period than the participants in SC group. After 8 weeks, the HbA1c level decreased significantly in the sRNT group (−0.72 ± 0.16%, *p* < 0.001) but not in the SC group (−0.35 ± 0.24%, *p* = 0.155); however, no difference was observed between the two groups. Similarly, BMI (−0.64 ± 0.11 kg/m^2^ vs. −0.60 ± 0.11 kg/m^2^) and waist circumference (−3.4 ± 0.7 cm vs. −2.5 ± 0.8 cm) decreased significantly over time in both groups, but the difference between the groups was not significant (Table 3). The level of triglycerides decreased significantly in sRNT group (−0.21 ± 0.08 mmol/L vs. 0.20 ± 0.17 mmol/L, *p* < 0.05) as compared to the SC group. HDL cholesterol (0.2 ± 0.0 mmol/L vs. 0.1 ± 0.1 mmol/L, ) increased in both groups whereas total cholesterol (−0.1 ± 0.2 mmol/L vs. −0.2 ± 0.2 mmol/L) improved in the sRNT group (*p* < 0.001) which was not seen in SC group (Table 3). 

### 3.3. Dietary Intake and Quality of Life (QoL)

Energy intake decreased in the sRNT group (−118 ± 50 kcal/day, *p* < 0.001) and increased in the SC group (20 ± 103 kcal/day, *p* < 0.05) (Table 3). Participants in the sRNT consumed less carbohydrate (%TEI −3.9% ± 1.2% vs. 2.2% ± 2.3%, *p* < 0.05), more protein (%TEI 0.9% ± 0.7% vs. −0.8% ± 0.8%, *p* < 0.05) and more fibre (2.15 ± 0.60 g/day vs. 0.60 ± 0.87 g/day, *p* < 0.05). The rate of underreported energy intake was comparable between the two groups, with an average rate of 84.3% at baseline. We assessed the adherence rates to DSF within the sRNT group, which increased significantly throughout the study period. The adherence rate of incorporating the DSF at least once daily during Ramadan achieved 89.4 ± 37.5%. The scores for satisfaction (−4.27 ± 0.96 vs. −2.13 ± 1.45) and QoL (−4.81 ± 1.55 vs. −0.74 ± 1.31) improved significantly in sRNT (*p* < 0.001) group as compared to the SC group. The worry and impact domains were comparable between the two groups (Table 3).

### 3.4. Self-monitoring Blood Glucose Profiles

Throughout Ramadan, participants in sRNT group had significantly better SMBG profiles at pre-dawn (6.9 mmol/L vs. 7.8 mmol/L, *p* < 0.05) and pre-bed (7.6 mmol/L vs. 8.6 mmol/L, *p* < 0.05) than the participants in SC group. After Ramadan, the SMBG profiles were consistently lower in participants of sRNT group than the SC group at each time point, but the difference was not significant (Figure 3). 

In total, 14 and 8 participants reported hypoglycemia during and after Ramadan, respectively, with 82% of them were on sulfonylurea medication. The majority of these participants were from the sRNT group (*n* = 11; 28.9%) vs. the SC group (*n* = 3; 13.6%), but the difference was not significant. The proportion of hypoglycemia improved after Ramadan in both groups but was still higher in sRNT (*n* = 6; 15.8%) than the SC (*n* = 2; 9.1%) group. However, the SC group experienced a significantly higher number of hypoglycemic episodes than those in the sRNT group both during (6.3 vs. 2.6, *p* < 0.05) and after Ramadan (3.0 vs. 1.3, *p* < 0.05). At the end of the study, about 21.1% (*n* = 8) of the participants in the sRNT and 9.1% (*n* = 2) in the SC group reduced the dosage of diabetes medications, while one participant (4.5%) from the SC group needed to increase the dosage of diabetes medications. 

### 3.5. Continuous Glucose Monitoring Profiles

The baseline characteristics of participants who participated in the CGM study (sRNT = 14, SC = 7) did not differ significantly from the whole study population. CGM participants (*n* = 21) and non-CGM participants (*n* = 43) were comparable in terms of age (49.3 ± 10.0 years vs. 47.8 ± 9.2 years), HbA1c level (7.6 ± 1.1% vs. 8.2 ± 1.7%), duration of diabetes (5.2 ± 4.1 years vs. 5.2 ± 4.1 years) and BMI (30.0 ± 6.2 kg/m^2^ vs. 30.4 ± 5.1 kg/m^2^) (Table 4). 

The participants from sRNT group had better average sensor glucose (6.85 mmol/L vs. 8.30 mmol/L, *p* < 0.05) and estimated HbA1c level (5.93% vs. 6.84%, *p* < 0.05) than those from SC group (Table 3). In addition, the participants from sRNT group spent significantly more time-in-target range (82.32% vs. 61.7%, *p* < 0.05) and spent less time-above-target range (16.61% vs. 36.21%, *p* < 0.05) than SC participants. No difference between the groups was observed for the time spent below-target-range (Table 4).

## 4. Discussion

The current study proved the feasibility of providing structured NT with incorporated DSF despite the fasting and feasting nature of Ramadan. Ramadan is time-sensitive, which limits the ability to comply with dietary changes in a short period [27]. Hence, patients’ preferences and choices are essential in optimizing dietary adherence. Although the randomized design eliminates selection bias [28], the random allocation may not match patients’ preference primarily related to DSF consumption, thereby compromising study validity. Indeed, the current study mimics real-life clinical experience where patients opt for their treatment choice. 

In this study, the number of participants in the sRNT and SC is markedly different, as more than half (63%) preferred to use the sRNT regimen, and 37% continued the standard nutrition care. The characteristics of the participants in both groups did not differ statistically. Still, the sRNT group had more women, longer duration of T2D, more comorbidities, and higher HbA1c levels than the participants in the SC group. It is understood that patients with long-standing T2D and more inadequate glycemic control tended to seek medical help and were willing to explore different treatment choices, which was particularly evident among women [29]. It is notable that women experienced a higher rate of hunger than men in early Ramadan but becoming less severe as Ramadan progresses [30]. Indeed, the hunger rating was comparable between men and women towards the end of Ramadan [30], suggesting a minimal effect of sex on the study outcomes. 

The fasting and feasting nature of Ramadan may cause a reduction in the adherence to DSF. Nonetheless, the present study achieved a good adherence rate of nearly 90% to DSF consumption during Ramadan, similar to the rate reported in another study using DSF in the non-Ramadan period [7]. This result indicates the feasibility of providing DSF as a part of the structured NT during Ramadan. Notably, the sRNT regimen was different from the other Ramadan-focused nutrition plans that did not include DSF in their intervention [11,12,14]. The incorporation of DSF in the sRNT was also different from other structured NT performed during the non-Ramadan period. In those studies, the DSF was used as a partial meal replacement for 1–2 meals/day to replace the usual meal [7,8,9]. In this study, DSF was consumed as part of the *Suhoor* and/or snack, together with other healthier meals within the prescribed carbohydrate and caloric limits.

Although the DSF was consumed with other healthy meals, it did not influence energy intake as the energy reduction was similar in both groups. Participants in sRNT group consumed more %TEI protein and fiber but less %TEI carbohydrates than the SC group, suggestive of the improvement in fasting blood glucose and triglycerides levels observed in this study. The improvement may be attributed to the macronutrient composition provided within the structured NT [9]. The DSF used in the study has distinctive characteristics as it comprises of low GI carbohydrates, rich in fiber, and high in whey protein [31]. In acute studies, the DSF produced lower postprandial glucose levels than breakfast cereals [31] or oats [32], which may explain the greater reduction in SMBG levels observed at pre-dawn and pre-bed times in the sRNT group as compared to the SC group. Additionally, participants of sRNT group had better CGM outcomes than the SC group. Although the CGM was not conducted in the entire study population, participants in sRNT spent almost 82% more time-in-target range, while it is recommended to spend 70% more time in glucose target range among people with diabetes [23]. Spending more time in glucose target range is critically needed as it reduces the progression to retinopathy, delays microalbuminuria development, and improves HbA1c levels [23]. The results are intriguing and strongly calls for replicating the study with bigger sample size.

The sRNT regimen resulted in a clinically significant glycemic reduction in patients with T2D. The incremental benefits are similar to those reported in the studies using structured NT delivered during the non-Ramadan period [7,8,9]. Although the reduction in HbA1c by 0.72% in sRNT group was not significantly different from the SC group (−0.35%), it was much higher than previously reported by Ramadan-focused education studies, where it ranged from 0.2%–0.4% among patients with T2D [10,11,12,13,14]. The improvement in HbA1c was unlikely due to medication or weight loss as both parameters remained unchanged throughout Ramadan. The improvement in glycemic parameters may have inadvertently caused a higher proportion of hypoglycemia in sRNT group. More participants in sRNT were taking sulphonylurea medications, whereas participants in SC received mainly metformin. It comes as no surprise that more participants in sRNT had hypoglycemia, owing to the nature of sulphonylureas that commonly causes hypoglycemia, especially during Ramadan [33]. No medication adjustment was made in either group prior to their enrollment in the study. Sulphonylureas may also induce weight gain [33], but we did not observe any significant difference regarding weight changes between the two groups.

The incidence of hypoglycemia is often associated with reduced QoL and increased fear of experiencing another hypoglycemic episode, which acts as a barrier for adherence to adequate treatment [34]. Despite a higher proportion of hypoglycemia in sRNT group, QoL and satisfaction with the treatment improved significantly, but this was not observed in SC patients. Compared to the usual care, a structured NT during Ramadan significantly reduced the number of hypoglycemic episodes, leading to more participants requiring a dosage reduction of diabetes medications in the sRNT than the SC group at the end of the study period. 

This study has some limitations. First, the patient preference design limits the ability to identify real differences between the groups, which is usually obtained in randomized controlled trials. However, the random allocation may not accord with patients’ preferences of nutrition therapy, especially related to the use of DSF during Ramadan, thereby compromising study validity. Understanding patients’ preferences replicate real-life routine practice, where the patients are given a choice of treatment. Second, this is a feasibility study, and it was not powered adequately. Nonetheless, we managed to detect significant benefits in clinical outcomes. Third, the high prevalence of underreported energy intake is a common issue as in other nutrition intervention studies, and it would have affected both the groups to the same extent. Lastly, the optimum repeat testing interval of 12 weeks is typically suggested for HbA1c, but the evidence supporting this recommendation is only based on expert opinion and studies with small sample size, including 9–10 patients [35,36]. Hirst et al. [37] demonstrated that 8 weeks’ change in HbA1c level following medication adjustments was reliable and robust in predicting the HbA1c change at 12 weeks [37], thereby suggesting that the 8-weeks duration may be sufficient to detect changes in HbA1c levels in our Ramadan study. The reduction in HbA1c at 8-weeks was also evident when using structured nutrition therapy in T2D and remained significant at 16-weeks [9]. Although Mottalib et al. [9] did not perform the study during Ramadan fasting, the results imply the potential of detecting changes in HbA1c up to 16 weeks following structured NT after Ramadan. 

## 5. Conclusions

The structured NT for Ramadan may be a feasible and beneficial program for T2D patients observing Ramadan fasting as it showed an improvement in some components of clinical outcomes and QoL. However, the adjustment in medications is required, especially for those receiving insulin secretagogues such as sulphonylureas, prior to adopting the structured NT that incorporated DSF. These results support the need for future clinical trials with adequate power and considering other parameters that might affect fasting metabolism, such as sleeping patterns. 

## Figures and Tables

**Figure 1 nutrients-12-00813-f001:**
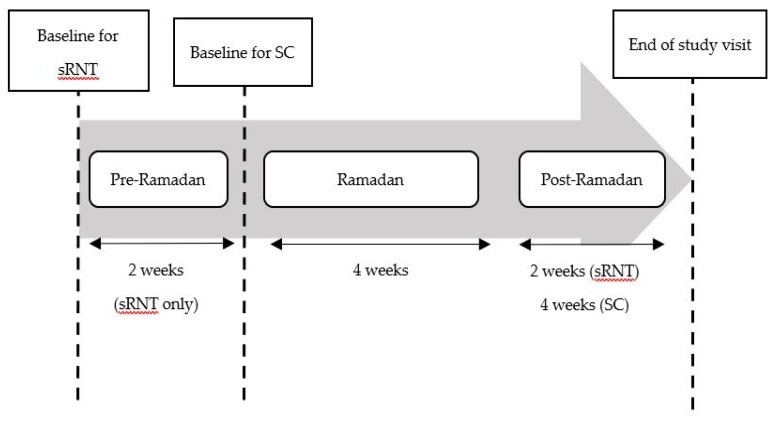
Study design.

**Figure 2 nutrients-12-00813-f002:**
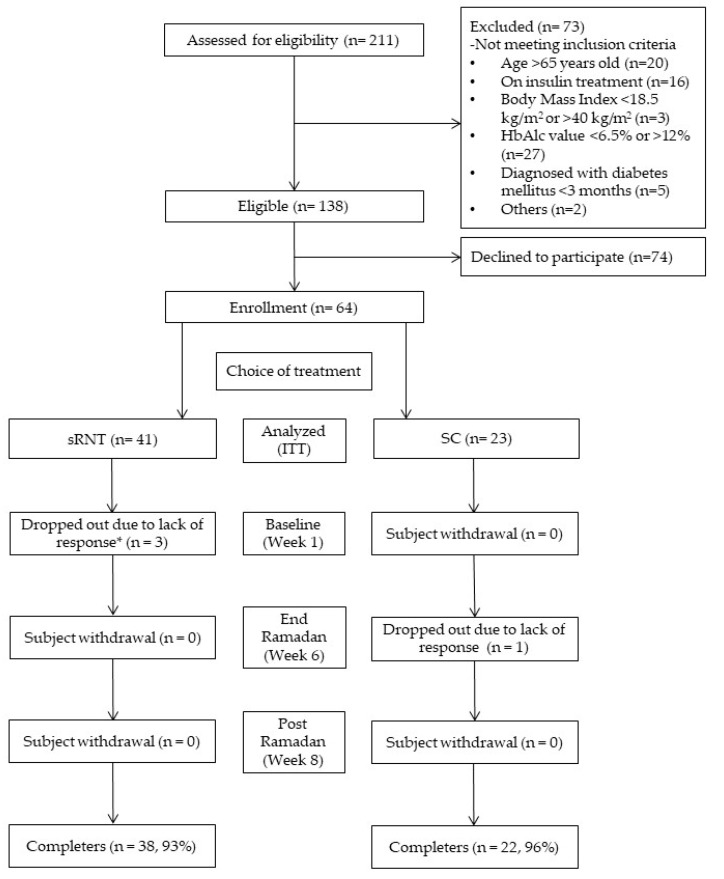
Screening and recruitment, * Three participants in structured Ramadan nutrition therapy, (sRNT) declined to participate before receiving the education. ITT, intention-to-treat analysis.

**Figure 3 nutrients-12-00813-f003:**
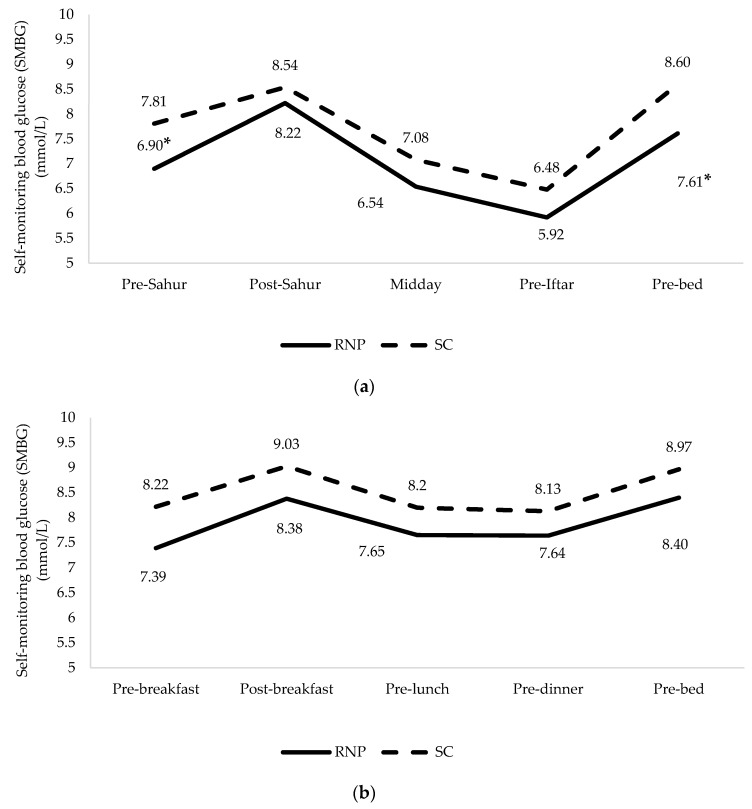
Self-monitoring blood glucose (SMBG) profiles between sRNT and SC, during. (**a**) Ramadan, (**b**) post-Ramadan, * Statistically significant differences between groups.

**Table 1 nutrients-12-00813-t001:** A summary of nutrition strategies in structured Ramadan Nutrition Plan and Standard Care.

Structured Ramadan Nutrition Therapy (sRNT)	Standard Care (SC)
Received structured Ramadan Nutrition Plan consisting of; pre-Ramadan nutrition education individualized energy and macronutrient prescriptions for Ramadan fasting consumption of diabetes-specific formula of at least 1 serving/day during *Suhoor* and/or pre-bed snack. the Ramadan toolkits (Ramadan flip chart, 14-day menu plan, Ramadan Nutrition Plate, and Festive season nutrition plan (*Syawal* nutrition plan).	Continued usual treatment consisting of; standard nutrition care recommendations [17] individualized modification of usual food intake suit Ramadan fasting The Healthy Malaysian plate [18]

**Table 2 nutrients-12-00813-t002:** Baseline characteristics of the participants (*n* = 64).

	sRNT (*n* = 41)		SC (*n* = 23)		*p*-Value
	*n* (%)	Mean ± SD	*n* (%)	Mean ± SD	
Age (years)		48 ± 9		48 ± 10	0.909
Male, *n* (%)	17 (41.5)		14 (60.9)		0.193
Duration of diabetes (years)		6 ± 4		4 ± 3	0.069
Number of co-morbidities		0.9 ± 0.8		0.8 ± 0.8	0.799
Number of oral diabetes medications		1.4 ± 0.6		1.4 ± 0.7	0.600
**Type of oral diabetes medications**					
Alpa-glucosidase inhibitors	2 (5.0)		1 (4.3)		
DPP4 inhibitors	6 (15.0)		2 (8.7)		
Metformin	12 (30.0)		12 (52.2)		
Sitagliptin plus metformin	17 (42.5)		7 (30.4)		
Sulphonylurea	19 (47.5)		9 (39.1)		
**Previous Ramadan experiences**					
Previous dietician encounter	19 (46.3)		12 (52.2)		0.795
Previous Ramadan advice	7 (17.1)		4 (17.4)		-
Hypoglycemia frequency	4 (9.8)		1 (4.3)		0.312
**Glucose-related outcomes**					
Fasting blood glucose (mmol/L)		7.82 ± 2.39		7.18 ± 2.26	0.304
HbA1c (%)		8.1 ± 1.7		7.8 ± 1.2	0.451
**Anthropometry**					
Weight (kg)		78.9 ± 18.2		76.2 ± 14.2	0.534
Body Mass Index (kg/m^2^)		30.9 ± 5.8		29.1 ± 4.6	0.201
**Waist circumference (cm)**		99.6 ± 12.7		94.9 ± 10.5	0.134
**Blood pressure**					
Systolic (mm Hg)		126 ± 16		130 ± 20	0.330
Diastolic (mm Hg)		86 ± 9		86 ± 11	0.970
**Lipid profiles**					
Total cholesterol (mmol/L)		4.91 ± 1.18		5.09 ± 1.60	0.599
Triglycerides (mmol/L)		1.73 ± 1.03		1.63 ± 0.90	0.694
HDL-cholesterol (mmol/L)		1.44 ± 0.31		1.36 ± 0.24	0.302
LDL- cholesterol (mmol/L)		2.66 ± 0.94		3.00 ± 1.41	0.302
**Energy intake and physical activity**					
Energy intake (kcal/day)		1472 ± 250		1429 ± 374	0.624
Total physical activity level (MET/minutes/weeks)		2564.83 ± 2424.44		2917.78 ± 2587.97	0.587
Sitting (MET/minutes/weeks)		2043.83 ± 909.64		2197.30 ± 798.24	0.502
**Quality of life**					
RV-DQOL13 Total score		29.12 ± 7.72		26.83 ± 5.86	0.220

HbA1c, glycated hemoglobin; HDL, High-density lipoprotein; LDL, Low-density lipoprotein; sRNT: Structured Ramadan Nutrition Therapy; SU, Standard Care.

**Table 3 nutrients-12-00813-t003:** Clinical outcomes, dietary intake, and quality of life throughout the study (*n* = 64).

Variables	*n*	Baseline	End-Ramadan Visit	Post-Ramadan Visit	Within-Group *p*-value	Changes at End-Ramadan from Baseline ^∆^	Changes at Post Ramadan from Baseline ^∆^	Interaction *p*-Value
**Fasting blood glucose (mmol/L)**								0.020
**sRNT**	41	7.82 ± 2.39	6.35 ± 1.64	6.93 ± 1.91	<0.001	−1.47 ± 0.30 *	−0.90 ± 0.32 *	
**SC**	23	7.18 ± 2.26	6.94 ± 1.61	7.34 ± 1.44	0.491	−0.25 ± 0.40	0.15 ± 0.30	
**HbA1c (%)**								0.174
**sRNT**	41	8.1 ± 1.7	Not measured	7.4 ± 1.3	<0.001	Not measured	−0.72 ± 0.16	
**SC**	23	7.8 ± 1.2		7.4 ± 1.2	0.155		−0.35 ± 0.24	
**Weight (kg)**								0.845
**sRNT**	41	78.9 ± 18.2	76.9 ± 18.0	77.3 ± 18.2	<0.001	−2.05 ± 0.24	−1.60 ± 0.28	
**SC**	23	76.2 ± 14.2	74.3 ± 13.9	74.6 ± 13.8	<0.001	−1.87 ± 0.32	−1.57 ± 0.31	
**Body Mass Index (kg/m^2^)**								0.772
**sRNT**	41	30.9 ± 5.8	30.1 ± 5.8	30.3 ± 5.8	<0.001	−0.80 ± 0.10	−0.64 ± 0.11	
**SC**	23	29.1 ± 4.6	28.4 ± 4.5	28.5 ± 4.4	<0.001	−0.71 ± 0.12	−0.60 ± 0.11	
**Waist circumference (cm)**								0.604
**sRNT**	41	99.6 ± 12.7	98.3 ± 13.2	96.2 ± 13.1	<0.001	−1.33 ± 0.63	−3.40 ± 0.65	
**SC**	23	94.9 ± 10.5	94.6 ± 11.4	92.4 ± 11.0	0.018	−0.30 ± 0.94	−2.47 ± 0.84	
**Systolic blood pressure (mmHg)**								0.886
**sRNT**	41	126 ± 16	121 ± 14	124 ± 10	0.140	−4.56 ± 2.43	−2.05 ± 2.32	
**SC**	23	130 ± 20	126 ± 15	127 ± 18	0.229	−4.35 ± 2.79	−3.48 ± 2.70	
**Diastolic blood pressure (mmHg)**								0.885
**sRNT**	41	86 ± 9	82 ± 10	84 ± 10	0.031	−3.49 ± 1.20	−1.02 ± 1.31	
**SC**	23	86 ± 11	82 ± 9	84 ± 10	0.097	−3.70 ± 1.46	−2.04 ± 1.89	
**Total cholesterol (mmol/L)**								0.641
**sRNT**	41	4.91 ± 1.18	4.35 ± 0.96	4.81 ± 0.94	<0.001	−0.55 ± 0.14	−0.10 ± 0.15	
**SC**	23	5.09 ± 1.60	4.64 ± 1.31	4.89 ± 1.35	0.135	−0.46 ± 0.25	−0.20 ± 0.24	
**Triglycerides (mmol/L)**								0.034
**sRNT**	41	1.73 ± 1.03	1.32 ± 0.57	1.52 ± 0.81	<0.001	−0.40 ± 0.10	−0.21 ± 0.08 *	
**SC**	23	1.63 ± 0.90	1.59 ±1.24	1.83 ± 1.28	0.296	−0.04 ± 0.18	0.20 ± 0.17	
**HDL-cholesterol (mmol/L)**								0.092
**sRNT**	41	1.44 ± 0.31	1.34 ± 0.25	1.60 ± 0.31	<0.001	−0.10 ± 0.02	0.16 ± 0.03	
**SC**	23	1.36 ± 0.24	1.32 ± 0.23	1.46 ± 0.32	0.013	−0.05 ± 0.04	0.10 ± 0.06	
**LDL-cholesterol (mmol/L)**								0.481
**sRNT**	40	2.66 ± 0.94†	2.44 ± 0.96	2.57 ± 0.89	0.093	−0.24 ± 0.12	−0.11 ± 0.11	
**SC**	23	3.00 ± 1.41	2.54 ± 0.97	2.53 ± 0.96	0.174	−0.33 ± 0.23	−0.34 ± 0.22	
**Dietary intake**								
**Energy intake (kcal/day)**								0.347
**sRNT**	41	1472 ± 250	1226 ± 229	1354 ± 317	<0.001	−246 ± 45	−118 ± 50	
**SC**	23	1429 ± 374	1234 ± 280	1449 ± 375	0.038	−195 ± 90	20 ± 103	
**Carbohydrate (%)**								0.015
**sRNT**	41	55 ± 5	53 ± 6	51 ± 6	0.003	−2.4 ± 1.1 *	−3.9 ± 1.2 *	
**SC**	23	51 ± 8	53 ± 7	53 ± 10	0.460	2.4 ± 2.0	2.2 ± 2.3	
**Protein (%)**								0.026
**sRNT**	41	15 ± 3	18 ± 3	16 ± 4	0.003	2.2 ± 0.6 *	0.9 ± 0.7	
**SC**	23	16 ± 3	16 ± 3	16 ± 3	0.569	−0.5 ± 0.8	−0.8 ± 0.8	
**Total fat (%)**								0.163
**sRNT**	41	30 ± 5	29 ± 5	32 ± 5	0.004	−0.5 ± 0.9	2.5 ± 0.9	
**SC**	23	33 ± 7	31 ± 5	32 ± 9	0.571	−2.0 ± 1.7	−1.1 ± 2.2	
**Total fiber intake (g/day)**								0.014
**sRNT**	41	5 ± 2	7 ± 2	7 ± 4	<0.001	2.79 ± 0.54	2.15 ± 0.60	
**SC**	23	4 ± 3	4 ± 2	5 ± 3	0.588	−0.01 ± 0.72	0.60 ± 0.87	
**Quality of life**								
**Satisfaction domain score**								0.279
**sRNT**	41	15.10 ± 3.59	11.24 ± 3.88	10.83 ± 5.49	<0.001	−3.85 ± 0.75	−4.27 ± 0.96	
**SC**	23	15.17 ± 3.97	13.26 ± 4.11	13.04 ± 4.83	0.199	−1.91 ± 0.76	−2.13 ± 1.45	
**Impact domain score**								0.113
**sRNT**	41	7.27 ± 3.75	6.56 ± 2.83	6.90 ± 2.78	0.396	−0.71 ± 0.57	−0.37 ± 0.66	
**SC**	23	5.78 ± 1.62	6.74 ± 2.20	6.83 ± 2.06	0.107	0.96 ± 0.52	1.04 ± 0.57	
**Worry domain score**								0.607
**sRNT**	41	6.76 ± 2.68	6.41 ± 2.72	6.59 ± 2.19	0.665	−0.34 ± 0.40	−0.17 ± 0.38	
**SC**	23	5.87 ± 2.07	5.96 ± 1.64	6.22 ± 2.24	0.466	0.09 ± 0.21	0.35 ± 0.36	
**Total score**								0.072
**sRNT**	41	29.12 ± 7.72	24.22 ± 6.83	24.32 ± 8.15	<0.001	−4.90 ± 1.25 *	−4.81 ± 1.55	
**SC**	23	26.83 ± 5.86	25.96 ± 6.62	26.09 ± 4.84	0.749	−0.87 ± 0.95	−0.74 ± 1.31	

Data are mean ± SD, * Significant difference between the group, *p* < 0.05, using Independent sample t-test, ^∆^ All data expressed as (mean ± SE) for absolute change in measures. HbA1c, glycated hemoglobin; HDL, High-density lipoprotein; LDL, Low-density lipoprotein; sRNT: Structured Ramadan Nutrition Therapy; SC, Standard Care.

**Table 4 nutrients-12-00813-t004:** Differences in continuous glucose monitoring reading during Ramadan. between sRNT and SC (*n* = 21).

Variables	Adjusted Mean [95% CI]		sRNT Versus SC Adjusted Mean Difference [95% CI]	*p*-Value
	sRNT (*n* = 14)	SC (*n* = 7)		
**Sensor glucose (mmol/L)**	6.85 [6.05; 7.65]	8.30 [7.17; 9.43]	−1.45 [−2.84; −0.07]	0.04
**Estimated A1c (%)**	5.93 [5.44; 6.42]	6.84 [6.14; 7.54]	−0.91 [−1.76; −0.05]	0.04
**Time in range**				
**Above range** **(>8.3 mmol/L (%))**	16.61 [7.91; 25.31]	36.21 [23.87; 48.55]	−19.60 [−34.76; −4.44]	0.01
**In target range** **(3.9–8.3 mmol/L (%))**	82.32 [73.88; 90.76]	61.79 [49.83; 73.75]	20.53 [5.86; 35.19]	0.01
**Below range** **(<3.9 mmol/L (%))**	1.45 [0.34; 2.56]	1.52 [−0.06; 3.10]	−0.07 [−2.02; 1.88]	0.94

sRNT: Structured Ramadan Nutrition Therapy; SC, Standard Care.

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
