# Peer review of "Comparison of Structured Nutrition Therapy for Ramadan with Standard Care in Type 2 Diabetes Patients"

_nutrients, 2020, doi:10.3390/nu12030813_

Round 1
Reviewer 1 Report
Yusof et al have conducted a human study to determine whether dietary modification could provide a more structured regimen that assists the clinical outcomes of individuals with type 2 diabetes undergoing fasting during Ramadan. The main outcome was a better glycemic metabolism as indicated by the fasting glucose level after 8 weeks on the modified diet. It was also claimed in the Abstract that there was a decrease in HbA1c levels but this was not the case using measured values rather than those calculated from a glucose monitoring device. A potential flaw was the use of an 8 week intervention period rather than 12 weeks which is recommended for HbA1c measurements. A citation was listed which was said to validate the 8 week period, however, reference 23 does not include an 8 week measurement and in this report no change in A1c was observed. Finally, even if the conclusions of the authors is correct, no information was provided regarding the meal planning or the nutrient composition. As such it does not enable the reader to evaluate the intervention against the outcomes.
Author Response
Point 1: Yusof et al have conducted a human study to determine whether dietary modification could provide a more structured regimen that assists the clinical outcomes of individuals with type 2 diabetes undergoing fasting during Ramadan. The main outcome was a better glycemic metabolism as indicated by the fasting glucose level after 8 weeks on the modified diet. It was also claimed in the Abstract that there was a decrease in HbA1c levels but this was not the case using measured values rather than those calculated from a glucose monitoring device.
General response: We appreciate all comments given by the reviewers as it improves the overall quality and readability of the manuscript.
Response 1: The differences in HbA1c reported in the abstract were within the group. We agreed this is maybe misleading and therefore we change the sentence by highlighting within and between the groups as follows:-
Line 34-35, page 1 (Abstract): Although not different between groups, HbA1c levels decreased significantly in sRNT (-0.72±0.16%, p<0.001) but not in SC group (-0.35±0.24%, p=0.155).
Yes, the HbA1c dropped significantly as estimated using the monitoring device. Nonetheless, the devices were only used by a sub-sample of the participants which may not represent the overall study population. Therefore, the results focus (in the Abstract section) on the metabolic parameters, not from the glucose monitoring device.
Point 2: A potential flaw was the use of an 8 week intervention period rather than 12 weeks which is recommended for HbA1c measurements. A citation was listed which was said to validate the 8 week period, however, reference 23 does not include an 8 week measurement and in this report no change in A1c was observed.
Response 2: Thank you for your comment. The right reference for this statement by right should be referred to the following, which was not numbered properly previously. We have correctly replaced the numbering to reference [37] to Hirst et al. 2014.
Hirst et al. (2014) explored how fast HbA1c changes after a change in glucose-lowering medication and identified that 12-week change in HbA1c can be predicted at 8 weeks after a medication change. Hirst’s study is important because the previous duration of HbA1c recommendation was mainly based on the expert opinion.
We modified the sentences accordingly to concur with the referred study too, in Discussion:
Line 408-415, page 14: Hirst et al. [37] demonstrated that 8 weeks’ change in HbA1c level following medication adjustments was reliable and robust in predicting the HbA1c change at 12 weeks [37]; thereby suggesting that the 8-weeks duration may be sufficient to detect changes in HbA1c levels in our Ramadan study. The reduction in HbA1c at 8-weeks was also evident when using structured nutrition therapy in T2D and remained significant at 16-weeks [9]. Although Mottalib et al [9] did not perform the study during Ramadan fasting, the results imply the potential of detecting changes in HbA1c following structured NT after Ramadan to predict HbA1c change at 12 weeks.
Point 3: Finally, even if the conclusions of the authors is correct, no information was provided regarding the meal planning or the nutrient composition. As such it does not enable the reader to evaluate the intervention against the outcomes
Response 3: We have added in the meal planning and nutrient composition to the Supplementary Files to enable reader to evaluate the intervention provided and may replicate them in future.
Table S1. Energy intake recommendation and carbohydrate exchange distribution
during Ramadan
Table S2. Daily energy intake distribution during Ramadan
Table S3. Sample menu for one day
Table S4. Ramadan nutrition plate composition
We have cited these tables in the text:
Line 108-110, page 3: Hence, a dietary plan for a reduced caloric intake of 1200-1500 kcal/day for women and 1500-1800 kcal/day for men was designed and distributed between the two main meals (Suhoor (pre-dawn meal) and Iftar (sunset meal) and two snacks (if necessary)) (Table S1).
Line 110-111, page 3: Also, dietary carbohydrates (~50% of total energy intake (TEI)) were distributed accordingly to the four mealtimes (Table S2).
Line 138-140, page 3-4: They were provided with a 14-day meal plan based on the Malaysian diet (Table S3) and Ramadan nutrition plate (Table S4).

Reviewer 2 Report
This is an interesting paper studying the positive effects of a structured nutrition therapy programme on various metabolic factors and quality of life in patients with type 2 diabetes during Ramadan. This is a novel study which has in my opinion some major and minor limitations.
Major:
- The major limitation is the non-randomised patients’ preference design which could affected the results significantly. Although the authors have mentioned this drawback in the limitations section I am only partly convinced why this study could not be performed as a randomized investigation. The sample size of the Intervention- and control groups are markedly different.
- How was the sample size/Power calculated? Although briefly written in the limitations Paragraph, similar studies with similar sample sizes should be at least cited.
- Table 2 shows the major results but in the intervention group only 38 were analysed not 41..
Minor:
- Please describe/present the ingredients of DSF
- In Fig.2 flow chart in the SC group 1 patient dropped out?? In the text but not shown here.
- In Table 1- would it be better to present the participants which were finally analysed, e.g. not including the drop outs?
- Triglycerides data should be presented in the text in 2 decimal places, e.g. f.ex. -0.21
Author Response
Major comments:
Point 1: The major limitation is the non-randomised patients’ preference design which could affected the results significantly. Although the authors have mentioned this drawback in the limitations section. I am only partly convinced why this study could not be performed as a randomized investigation. The sample size of the Intervention- and control groups are markedly different.
General response: We appreciate all comments given by the reviewers as it improves the overall quality and readability of the manuscript.
Response 1: We have edited the opening part of our Discussion to address this comment:
Line 332-348, page 12-13: The current study proved the feasibility of providing structured NT that incorporated DSF despite the fasting and feasting nature of Ramadan. Ramadan is time-sensitive, which limits the ability to comply with dietary changes in a short period of time [27]. Hence, patients’ preference and choice are essential in optimizing dietary adherence. Although the randomized design eliminates selection bias [28], the random allocation may not accord with patients’ preference especially related to DSF consumption thereby compromising study validity. Indeed, the current study mimics real-life clinical experience where patients opt for their treatment choice.
In this study, the number of participants in the sRNT and SC are markedly different. More than half (63%) of the participants preferred to use sRNT regimen and 37% continued the standard nutrition care. The characteristics of the participants in both groups did not differ statistically, but the sRNT group had more women, longer duration of T2D, more comorbidities, and higher HbA1c levels than the participants in SC group. It is understood that patients with long-standing T2D and poorer glycaemic control tended to seek medical help and were willing to explore different treatment choices, which was particularly evident among women [29]. It is notable that women experienced higher rating of hunger than men at early Ramadan but becoming less severe as Ramadan progresses [30]. Indeed, the hunger rating was comparable between men and women towards the end of Ramadan [30] suggesting minimal effect of sex on the study outcomes.
Point 2: How was the sample size/Power calculated? Although briefly written in the limitations Paragraph, similar studies with similar sample sizes should be at least cited.
Response 2: We have included our sample size calculation in Section 2.5 Sample Size
Line 201-206, page 5: A total of 70 participants was sufficient to detect a mean difference of 0.84% in HbA1c level among T2DM patients after following a tele-monitoring intervention during Ramadan with 80% power and 90% confidence interval [24]. Considering dropout rate of 20%, a total of 84 participants would be needed in the study. In this feasibility study, we recruited at least 70% participants from the primary planned statistical analyses [25]. Hence, a minimum of 59 subjects were required for this study.
Point 3: Table 2 shows the major results but in the intervention group only 38 were analysed not 41..
Response 3: Thank you for your comment. We have previously explained in Line 215-220, page 5 that the analyses was performed using the ITT. Results were similar when the analyses was performed using the PP methods. We modified as the following to improve overall explanation. We also justified the reason of performing the ITT instead of PP analysis for the study.
Line 215-220, page 5: The primary analysis was intention-to-treat (ITT). All participants who are enrolled were included in the analysis to allow us to draw accurate conclusion regarding the effectiveness of the intervention regardless of drop-out and adherence to study protocol. ITT analysis should always be considered as the ideal primary analysis as it provides an unbiased estimate of treatment effect [26]. Imputation of missing data using the last observation carried forward produced no effect on the significance in per-protocol (PP) analysis. Hence, we present the results for the planned ITT analysis.
Minor comments
Point 4: Please describe/present the ingredients of DS
Response 4: We have included in Method Section 2.3.1 Structured Ramadan Nutrition Therapy Group
Line 134-136, page 3: The DSF provides approximately 250 kcal per serving with balanced macronutrients composition. The DSF contains prebiotics, is rich in dietary fibre, high in whey protein and low in glycemic index (GI) (Nutren Untuk Diabetik®, Nestlé Health Science, Switzerland).
Point 5: In Fig.2 flow chart in the SC group 1 patient dropped out?? In the text but not shown here.
Response 5: Thank you, we have corrected Fig. 2 (page 6, highlighted). 1 patient in SC group dropped out at baseline (Week 1).
Point 6: In Table 1- would it be better to present the participants which were finally analysed, e.g. not including the drop outs?
Response 6: Thank you. In this paper we focused on ITT analysis, as previously explained in Response 3 (see above).
Point 7: Triglycerides data should be presented in the text in 2 decimal places, e.g. f.ex. -0.21
Response 3: Thank you, changes have been made accordingly
Line 32, page 1 (Abstract): 0.21 ± 0.81 mmol/L vs 0.20 ± 0.17 mmol/L, p<0.05
Line 265-267, page 8: The level of triglycerides reduced significantly in sRNT group (-0.21 ± 0.081 mmol/L vs. 0.20 ± 0.172 mmol/L, p<0.05) as compared to the SC group

Reviewer 3 Report
The Comparison of Structured Nutrition therapy for Ramadan with Standard Care in Type 2 Diabetes Patients investigates an important topic testing as a novelty the Ramadan Nutrition Plan as proposed by the International Diabetes Federation and the Diabetes and Ramadan International Alliance. However, the manuscript could be improved by a more in depth discussion of the results in relation to previous study protocols and by providing some clarifications:
GENERAL COMMENTS
The text would benefit from English language editing.
INTRODUCTION
- Please provide a more in depth description of the pathophysiology in Ramadan fasting type 2 Diabetes patients. Furthermore, previous studies about this topic should be described more in detail and the reference list should be extended as there are some important studies such as CREED or READ that are not covered.
MATERIAL & METHODS
- Study Population
a) Information about relevant baseline characteristics are missing such as previous hypo/hyperglycemia frequency, activity levels, who prepares meals (patient, relatives?)
- Study Design
a) Please describe and provide more details about the two different nutritional regimens. How was the adherence to the protocol?
b) The study design considered patient's preference for one study arm. This might be a relevant bias; especially as there were more women in the intervention group. Who was preparing the meals? Please discuss a possible bias with regard to this issue. Were relatives responsible for the preparation of food also instructed?
c) How were the dietary intakes and macronutrients assessed? The origin of the results in table 2 is not well described: how have they been assessed and how reliable, therefore, is a comparison of the two study groups?.
d) Outcome measurements
Sleeping pattern and acitivity levels were not assessed but might have an impact on metabolism. Please comment this possible bias or provide if possible more data.
DISCUSSION
1) As listed above the issue of more women in the intervention group should be also discussed in this section. What have previous studies shown with regard to fasting metabolism in both genders, which difference have been detected and might, therefore, also have affected the presented study results?
2) The two groups are different with regard to oral diabetes medications. Please discuss possible effects on hypoglycaemia/hyperglycemia tendency or on the results (weight loss) respectively.
3) DSF was incorporated in a different way than in previous studies. Please discuss how this might have affected the presented results, also in comparison to previous use of DSF as meal replacement.
CONCLUSIONS
Please include in your conclusion that these results support a further trial including a higher number of patients with adequate power calculation.
Author Response
General comment
Point 1: The text would benefit from English language editing.
Response 1: We have previously employed the service of a certified proof-reader (Editage). Hereby, we attach the receipt and a letter from the editor of Editage.
Point 2: INTRODUCTION
- Please provide a more in depth description of the pathophysiology in Ramadan fasting type 2 Diabetes patients. Furthermore, previous studies about this topic should be described more in detail and the reference list should be extended as there are some important studies such as CREED or READ that are not covered.
Response 2: We have added the physiology of Ramadan fasting with the focus on individuals with diabetes. Data from CREED [3] and READ [4] studies were included in the introduction. Therefore, the whole section have been revised accordingly as follow:-
Line 51-59, para 2, page 2: The shift has significant implications for physiology process, in particular to people with diabetes. When fasting, insulin resistance or deficiency state can cause excessive glycogen breakdown lead to gluconeogenesis. The alteration put individuals with diabetes at heightened risk to develop hypoglycaemia, hyperglycaemia, ketoacidosis, and dehydration [2]. Therefore, they are exempted from fasting but many Muslim adults with diabetes continue to observe Ramadan fasting, even against medical advice. The CREED study documented that 94% of the patients with type 2 diabetes (T2D) fasted for at least 15 days during Ramadan, while more than half (64%) fasting everyday [3]. Nonetheless, with a proper Ramadan-focused education, the medical risk including hypoglycemia can be minimized [4].
Point 3: MATERIAL & METHODS
- Study Population a) Information about relevant baseline characteristics are missing such as previous hypo/hyperglycemiafrequency, activity levels, who prepares meals (patient, relatives?)
Response 3.1:
- We have added number of participants in sRNT vs SC group who had hypoglycaemia during the previous Ramadan in Table 1. However, no information about previous Ramadan hypoglycaemia nor hyperglycaemia frequency was available.
- For activity level, total physical activity level (MET/minutes/weeks) and sitting (MET/minutes/weeks) have already been presented on Table 1.
- The patient themselves prepares meals, as they are adults who are able to perform activities of daily living (ADLs) independently, such as cooking. We have also added in Method Section 2.1 Study Population:
Line 84-85, page 2: They were also adults who were able to perform activities of daily living (ADLs) independently, such as cooking and preparing meals themselves.
2. Study Design a) Please describe and provide more details about the two different nutritional regimens. How was the adherence to the protocol?
Response 3.2a: The description about nutrition strategies of the two group are summarized in Table 1 (page 4) as below. We have also added the energy prescription, macronutrient distribution and sample menu in supplementary files (Table S1-S4).
Table 1: A summary of nutrition strategies in structured Ramadan Nutrition Plan and Standard care
Structured Ramadan Nutrition Therapy (sRNT) |
Standard Care |
Received structured Ramadan Nutrition Plan consisting of;
|
Continued usual treatment consisting of;
|
The adherence to protocol relied on dietary intake of participants. The proportion of carbohydrate, protein and fat intake were within the recommendations in both groups. Within the sRNT, we assessed the adherence rates to DSF, which increased significantly throughout the study period. It was mentioned in Discussion, Line 350-353, page 13: Nonetheless, the present study achieved a good adherence rate of nearly 90% to DSF consumption during Ramadan, and it is comparable to the rate reported in another study using DSF in the non-Ramadan period [7]. This result indicates the feasibility of providing DSF as a part of the structured NT during Ramadan.
b) The study design considered patient's preference for one study arm. This might be a relevant bias; especially as there were more women in the intervention group. Who was preparing the meals? Please discuss a possible bias with regard to this issue. Were relatives responsible for the preparation of food also instructed?
Response 3.2b: It is undeniable that considering patient’s preference to one study arm incurred selection bias especially more women opted to be in the intervention group. We started a discussion with this issue as the following:-
Line 332-348, page 12-13: The current study proved the feasibility of providing structured NT that incorporated DSF despite the fasting and feasting nature of Ramadan. Ramadan is time-sensitive, which limits the ability to comply with dietary changes in a short period of time [27]. Hence, patients’ preference and choice are essential in optimizing dietary adherence. Although the randomized design eliminates selection bias [28], the random allocation may not accord with patients’ preference especially related to DSF consumption thereby compromising study validity. Indeed, the current study mimics real-life clinical experience where patients opt for their treatment choice.
In this study, the number of participants in the sRNT and SC are markedly different. More than half (63%) of the participants preferred to use sRNT regimen and 37% continued the standard nutrition care. The characteristics of the participants in both groups did not differ statistically, but the sRNT group had more women, longer duration of T2D, more comorbidities, and higher HbA1c levels than the participants in SC group. It is understood that patients with long-standing T2D and poorer glycaemic control tended to seek medical help and were willing to explore different treatment choices, which was particularly evident among women [29]. It is notable that women experienced higher rating of hunger than men at early Ramadan but becoming less severe as Ramadan progresses [30]. Indeed, the hunger rating was comparable between men and women towards the end of Ramadan [30] suggesting minimal effect of sex on the study outcomes.
c) How were the dietary intakes and macronutrients assessed? The origin of the results in table 2 is not well described: how have they been assessed and how reliable, therefore, is a comparison of the two studygroups?.
Response 3.2c: Thank you, we have addressed this comment in Section 2.4 Outcome Measurements
Line 170-177, page 4: Dietary intake and macronutrients were assessed using 3-day food records. Participants were taught and written instructions were given for recording the food intake. The research dietician reviewed food records at each visit. The average of 3-day food intake data were analysed for daily total energy intake (TEI), the proportion of macronutrient composition from TEI (%TEI), total fibre and selected micronutrients intake using the Nutritionist Pro software (V.5.1.0, Axxya Systems, WA, USA). The reliability of food intake data was ensured by including only those with plausible energy intake of 400-3500 kcal/day for women, and 700- 4000 kcal/day [19, 20]. The assessment of under-reporting of energy intake were performed at baseline using the Goldberg method [21].
As a note, plausible energy intake of 500-3500 kcal/day for women and 800-4000 kcal/day for men has been reported [19]. However, a study by Mafauzy et al [20] found that there was a decrease in energy intake (103 kcal/day). Hence, we allowed the lower limit of energy intake to be 400 kcal/day for women and 700 kcal/day for men in our study. No respondents were excluded from dietary analysis based on this cut-off point.
We acknowledged the high prevalence of underreporting among the study subjects with no different between group which we believe would impact to the same extent in both groups. This information was mentioned in study limitation:
Line 404-406, page 14: Third, the high prevalence of underreported energy intake is a common issue in other nutritional intervention studies, but in our study, it would have affected both the groups to the same extent.
d) Outcome measurements
Sleeping pattern and acitivity levels were not assessed but might have an impact on metabolism. Please comment this possible bias or provide if possible more data.
Response 3.2d: Physical activity level was measured using IPAQ at baseline of the study, which generally all subjects spent more time in sedentary activities with no difference between groups. We did not assess sleeping pattern and we agreed the sleeping pattern influenced blood glucose level which warrants future studies; this point has been added in Conclusion (Line 421-423, page 14).
Point 4: DISCUSSION
- As listed above the issue of more women in the intervention group should be also discussed in this section. What have previous studies shown with regard to fasting metabolism in both genders, which difference have been detected and might, therefore, also have affected the presented study results?
Response 4.1: We have discussed in terms of health seeking behaviour especially among women. It is notable that women experienced a higher rate of hunger than men at early Ramadan but becoming less severe as Ramadan progresses [30]. Indeed, the hunger rating was comparable between men and women towards the end of Ramadan [30] suggesting minimal effect of sex on the study outcomes. This was mentioned in Line 345-348, page 13.
2. The two groups are different with regard to oral diabetes medications. Please discuss possible effects on hypoglycaemia/hyperglycemia tendency or on the results (weight loss) respectively.
Response 4.2: We agree that there may be possible effects on hypoglycemia tendency as more participants in sRNT received sulphonylureas. We have added this response in Discussion:
Line 384-389, page 13-14: More participants in sRNT were on sulphonylureas whereas participants in SC received mainly metformin. It comes as no surprise that more participants in sRNT had hypoglycemia, owing to the nature of sulphonylureas that commonly causes hypoglycemia especially during Ramadan [33]. No medication adjustment was made in either group prior to their enrollment in the study. Sulphonylureas may also induce weight gain [33] but in our study, we observed no significant difference regarding weight changes between the two groups.
3. DSF was incorporated in a different way than in previous studies. Please discuss how this might have affected the presented results, also in comparison to previous use of DSF as meal replacement.
Response 4.3: We have added this response in Discussion:
Line 349-375, page 13: The best criterion for inclusion of DSF during Ramadan is uncertain. Besides, the fasting and feasting nature of Ramadan may cause reduction in the adherence to DSF. Nonetheless, the present study achieved a good adherence rate of nearly 90% to DSF consumption during Ramadan, and it is comparable to the rate reported in another study using DSF in the non-Ramadan period [7]. This result indicates the feasibility of providing DSF as a part of the structured NT during Ramadan. Also, the sRNT regimen was different from the other Ramadan-focused nutrition plans as other studies did not include DSF in their intervention [11-12, 14] The incorporation of DSF in sRNT was also different from other structured NT that was performed during the non-Ramadan period, wherein the DSF was used as a partial meal replacement for 1-2 meals/day to replace the usual meal [7-9]. In this study, the DSF was consumed as part of the Suhoor and/or snacks together with other healthy meal within the prescribed carbohydrate and caloric limits.
Although the DSF was consumed with other healthy meals, it did not influence energy intake as the energy reduction was similar in both groups. Participants in sRNT group consumed more %TEI protein and fibre but less %TEI carbohydrates than the SC group during the study period, which is suggestive of the improvement in fasting blood glucose and triglycerides levels observed in this study. The improvement may be attributed to the macronutrient composition provided within the structured NT [9]. The DSF has distinctive characteristics as it comprises of low GI carbohydrates, rich in fibre, and high in whey protein [31]. In acute studies, the DSF produced lower postprandial glucose levels than breakfast cereals [31] or oats [32], which may explain the greater reduction in SMBG levels observed at pre-dawn and pre-bed times in the sRNT group as compared to the SC group. Additionally, participants of sRNT group had better CGM outcomes than the SC group. Although the CGM was not conducted in the entire study population, participants in sRNT spent almost 82% more time-in-target range. The results are intriguing and strongly calls for replicating the study with bigger sample size. It is recommended to spend 70% more time in glucose target range among people with diabetes [23]. Spending more time in glucose target range is critically needed as it reduced the progression to retinopathy, delayed microalbuminuria development, and improved HbA1c levels [23].
Point 5: CONCLUSIONS
Please include in your conclusion that these results support a further trial including a higher number of patients with adequate power calculation.
Response 5:
Line 421-423, page y: These results support the need for future clinical trials with a higher number of patients, adequate power calculation, and considering other parameters that might affect fasting metabolism such as sleeping pattern.

Round 2
Reviewer 1 Report
The authors have significantly improved the content of their manuscript.